# Structural Diversity of Ubiquitin E3 Ligase

**DOI:** 10.3390/molecules26216682

**Published:** 2021-11-04

**Authors:** Sachiko Toma-Fukai, Toshiyuki Shimizu

**Affiliations:** 1Graduate School of Science and Technology, Nara Institute of Science and Technology, 8916-5 Takayama-cho, Ikoma 630-0192, Japan; 2Graduate School of Pharmaceutical Sciences, The University of Tokyo, 7-3-1 Hongo, Bunkyo-ku, Tokyo 113-0033, Japan

**Keywords:** post-translational modification, ubiquitin E3 ligase, structural biology, X-ray crystallography

## Abstract

The post-translational modification of proteins regulates many biological processes. Their dysfunction relates to diseases. Ubiquitination is one of the post-translational modifications that target lysine residue and regulate many cellular processes. Three enzymes are required for achieving the ubiquitination reaction: ubiquitin-activating enzyme (E1), ubiquitin-conjugating enzyme (E2), and ubiquitin ligase (E3). E3s play a pivotal role in selecting substrates. Many structural studies have been conducted to reveal the molecular mechanism of the ubiquitination reaction. Recently, the structure of PCAF_N, a newly categorized E3 ligase, was reported. We present a review of the recent progress toward the structural understanding of E3 ligases.

## 1. Introduction

Ubiquitination (also known as ubiquitylation) is one of the post-translational modifications and most widely used. Ubiquitination targets lysine residue and regulates many cellular processes, for example, protein degradation, DNA repair, and signal transduction.

The most well-known function of ubiquitination is the selective protein degradation of proteins. The early studies revealed that ubiquitination mediates degradation by the 26S proteasome [1,2]. Consecutive studies have uncovered the various roles of ubiquitination that regulate a multitude of cellular functions. The modification of ubiquitination is analogous to protein phosphorylation that is reversibly reaction-regulated by deubiquitination enzymes in a similar way to phosphatases. Different ubiquitination patterns recognized by a specific effector protein transduce a different downstream signal. Ubiquitination and a modification by the ubiquitin-like protein are achieved by three enzymes: a ubiquitin-activating enzyme (E1), a ubiquitin-conjugating enzyme (E2), and ubiquitin ligase (E3) [3,4,5] (Figure 1A). It is thought that E3 is responsible for diverse ubiquitination patterns in a cell among three enzymes. The biological functions relating to ubiquitin and the ubiquitin-like protein have not been fully elucidated yet.

E3 proteins have emerged as pivotal targets for drug discovery using the function of targeted protein degradation. One of the most attractive approaches is proteolysis-targeting chimera (PROTAC). This new technology, PROTAC, can induce the target protein degradation [6]. The ubiquitin-proteasome system is used for destructing damaged proteins or proteins no longer required in the cell [7,8]. A PROTAC has two functional ligands connected by a linker: one binds to a target protein and the other binds to an E3 ligase. The functional ligand is developed based on the peptide or small molecule [9]. To date, the PROTAC technology is used to target varieties of proteins, including transcription factors, skeleton proteins, enzymes, and regulatory proteins. This technology is expected to give one promising way for developing drugs on undruggable proteins [10,11,12].

The E3 protein is a remarkable enzyme for understanding the ubiquitination system in a cell and for developing a drug. Structural knowledge is indispensable to establish the molecular basis of the ubiquitination mechanism. Many structural biology studies have been reported. Therefore, many excellent structural reviews focused on the ubiquitination system have been published. In this review, we briefly introduce previous structural biology studies of ubiquitin and ubiquitin-like proteins and also describe the new categorized E3 protein family. 

## 2. Ubiquitination

### 2.1. Structure of Ubiquitin

Ubiquitin (ubiquitous immunopoietic polypeptide or UBIP) was first found in the bovine thymus during isolation of the thymic polypeptide hormone thymopoietin, and its amino acid sequence was reported in 1975 [13]. Ubiquitin consists of 76 amino acids and the first structure was determined at 1.8 Å by X-ray crystallography in 1987 (PDB ID: 1ubq) [14]. The canonical ubiquitin fold is formed by a 5-stranded β-sheet, a short 3_10_ helix, and a 3.5-turn α-helix. The carboxy-terminal tail of ubiquitin is exposed to a solvent that allows its covalent linkage to target proteins (Figure 1B). 

### 2.2. Ubiquitination Reaction

A ubiquitination reaction is achieved by three enzymes [3,4,5] (Figure 1A). An E1 catalyzes the formation of a covalent thioester bond between the catalytic cysteine and the di-glycine motif on the C-terminus of ubiquitin by a magnesium ion and ATP. The E1 transfers ubiquitin to the catalytic cysteine of an E2 to form an E2~ubiquitin thioester complex (~indicates a thioester bond). An E3 binds E2~ubiquitin and the substrate to facilitate isopeptide bond formation between the C-terminal carboxyl of ubiquitin and the ε-amino group of a lysine side chain or free N-terminal amino group of the substrate. E3 ligases work for recruiting substrates and facilitating the transfer of ubiquitin from an E2-conjugating enzyme to the target protein.

The number of enzymes involving ubiquitination is increasing. It was reported that the human genome encodes two E1s, approximately 38 E2s, and more than 600 E3s [15,16,17]. There is a much larger number of E3s among ubiquitination enzymes, indicating that E3s are the key enzymes contributing to diverse ubiquitination functions [18]. The structural details of these enzymes are described below.

### 2.3. Chain Diversity and Their Function

All seven Lys residues on the ubiquitin molecule are ubiquitinated. In addition to the Lys residues, the N-terminal amino group of the first Met is also ubiquitinated. Substrate proteins can be modified at single or multiple Lys residues with either a single ubiquitin molecule (mono- and multi-monoubiquitylation, respectively) or ubiquitin polymers (polyubiquitylation). The ubiquitin chain comprised of only a single linkage type is called homotypic. On the other hand, heterotypic chains consist of mixed linkages within the same polymer [19]. The branched ubiquitin chain was also observed [20]. Additionally, the ubiquitin molecule is modified by other post-translational modifications: acetylation [21], phosphorylation [22,23,24,25,26], and sumoylation [27,28,29,30]. 

Proteomics analysis showed that all linkage types exist in the cell [31,32,33,34,35]. Among them, Lys48-linked chains that bring the substrate protein to the proteasome are the predominant chain linkage. The second abundant modification is the Lys63-linked chain that has various nondegradative functions in the cell: endocytic pathway [35] or regulating kinase activation in the NF-κB signaling pathway, and so on [36]. Linkage-specific hydrolases and proteins were found. These findings revealed that molecules in charge of writer, reader, and eraser exist in the ubiquitination system. 

**Figure 1 molecules-26-06682-f001:**
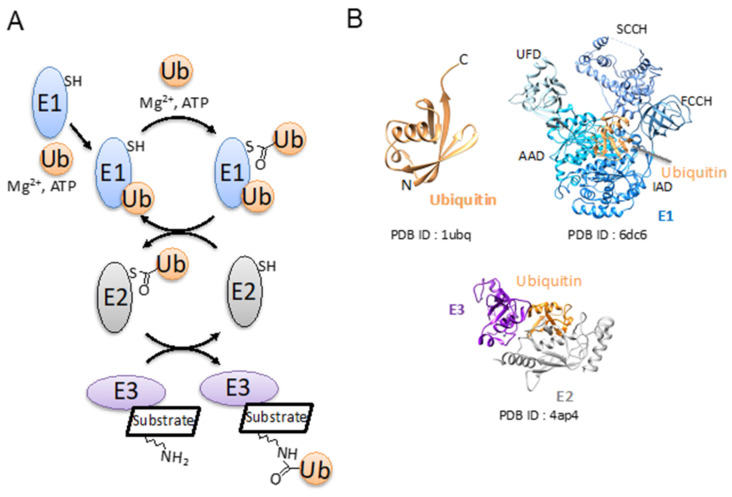
Schematic diagram of the ubiquitin conjugation system and the crystal structures of ubiquitin, E1, E2, and E3. (**A**) Schematic diagram of ubiquitin conjugation system. All molecules (ubiquitin, E1, E2, and E3) relating to the system are drawn in a circle and colored in orange, blue, gray, and purple, respectively. (**B**) The crystal structures of ubiquitin, E1, and E2 are drawn in a ribbon diagram and colored in orange, blue, and gray, respectively. PDB IDs are indicated under each structure.

## 3. Structural Study of Ubiquitin and Ubiquitin Ligases

### 3.1. Structure of E1

E1 is the initiator of ubiquitination. E1 activates ubiquitin in a two-step reaction. In the first step, E1 adenylates the C-terminal of ubiquitin. The second step is the formation of a thioester bond between the Ub C-terminal and the E1 catalytic cysteine [37,38]. After that, the E1~Ub intermediate recruits an E2 to facilitate the thioester transfer of Ub from the E1 to E2 catalytic cysteine: an E1~Ub complex places the E2 catalytic cysteine in proximity of the E1~Ub thioester bond [39,40] (Figure 1A).

In addition to ubiquitin, there are 16 additional ubiquitin-like proteins (Ubls). They are activated by cognate E1s [41,42]. Therefore, E1 is the first gatekeeper of the ubiquitin cascade. E1 and its complex structures with the ubiquitin molecule and E2 enzyme have been reported [43,44,45,46,47,48]. The first determined structure was *Saccharomyces cerevisiae* E1 (ubiquitin-activating enzyme, known as Uba1) in complex with Ub. Interestingly, the ubiquitin system is unique among Ub-like modifiers because it has 10–40 different E2s having distinct functions [49,50]. One or two kinds of E1 must activate all E2s. Structural studies of E1 revealed both E1 catalytic activity and how to activate several E2 enzymes promiscuously. 

Uba1 harbors seven domains and achieves dual catalytic activities by large conformational changes of the domains (Figure 1B). The inactive domain (IAD) and active adenylation domain (AAD) assemble into a pseudodimer. The AAD binds Ub, ATP, and Mg^2+^ and carries out the adenylation reaction. The cysteine domain is divided into two catalytic half-domains in which the first domain is FCCH and the second one is SCCH. The catalytic cysteine is located on the SCCH domain [43]. Studies of E1 for SUMO have shown that the SCCH domain rotates ~130° after the adenylation. This structural transition relocates the catalytic cysteine, and other structural elements are required for catalysis of the thioester bond formation into the active site [51,52]. A similar structural transition was observed in the S. pombe Uba1 in which the SCCH domain rotated 106°. The ubiquitin fold domain (UFD) recruits E2 enzymes and places the catalytic cysteines of E1 and E2 into proximity to facilitate E1-E2 thioester transfer [44].

The basic motif at the N-terminal helix of the core of E2s is demonstrated to be important for Ubc2 binding to Uba1 [53]. However, both the predicted E1 interaction region and the basic motif itself are not highly conserved. These pieces of evidence suggest that sequence and structural plasticity at the Uba1-E2 interface might underlie the promiscuity of Uba1 and variability in the affinities of E1-E2 interactions [54]. For a long time, a mammalian E1 structure has been unknown. Recently, it was reported and demonstrated that in the human Uba1-Ub structure, Uba1 shares a conserved overall structure and mechanism, indicating that structural plasticity as a mechanism that underlies promiscuity in E1–E2 interactions are a conserved element of the human ubiquitin E1-E2 system [54].

### 3.2. Structure of E2

The E2 family comprises 40 genes in humans. Human E2s are classified into 17 subfamilies based on phylogenetic analyses. NMR and small-angle X-ray scattering (SAXS) analyses revealed that the structure of UbcH5c~ubiquitin is highly dynamic and prefers to adopt open conformations without E3 (Figure 2A) [55,56].

In 1992, the first crystal structure of E2 was UBC of *Arabidopsis thaliana* [57]. E2s harbor a core ubiquitin-conjugating (UBC) domain harboring the catalytic Cys residue. The UBC domain consists of four α-helices and an antiparallel β-sheet comprised of four β-strands. E2s are highly homologous in sequence and have a well-conserved structure in which the catalytic Cys residue is located on the shallow groove formed by a short loop between α-helix2 and α-helix3. E1 interacts with α-helix1, and E3 interacts with loop1 and loop2 (Figure 2B). 

### 3.3. Structure of E3

#### 3.3.1. Protein Family of E3

Ubiquitin modification induces diverse cellular signals. Therefore, precise substrate selection ensures accurate ubiquitin signaling. E3 is responsible for substrate selection. The number of genes encoding E3 should be the largest among the three enzymes, indicating that the diversity of the E3 protein provides the precise substrate selection. More than 600 E3s are encoded in the human genome and exhibit structural diversity, in contrast to E1 and E2. Classically, these E3s are classified into three protein families, RING (interesting new gene), HECT (homologous to E6AP C-terminus), and RBR (RING-between-RING). Based on structural and biochemical studies, three families have two main types of ubiquitin transfer mechanisms. RING E3s catalyze the direct transfer of ubiquitin from E2~ubiquitin to the substrate. By contrast, HECT and RBR E3s harbor a catalytic cysteine that receives ubiquitin from E2~ubiquitin, and then transfers this ubiquitin to the substrate (Figure 3A). In addition to these three families, the crystal structure of the newly categorized E3 domain named PCAF_N was reported recently [18]. 

In this section, we introduce the structural studies of three classical E3 families and atypical E3s for UBL.

#### 3.3.2. Ring 

Among the three classical E3s, the RING domain is smallest and has a distinct transfer mechanism. The RING domain bears two zinc ions. The Zn coordination arranged in a cross-braced configuration is essential for RING domain folding (Figure 3A). Though the RING domain is a small and simple architecture, RING E3s exert their E3 activity with a highly diverse quaternary architecture [55] (Table 1). Some RING E3s exhibit fully E3 activity as monomers, such as in CBL [58]. Other RING domains are active as oligomers. For example, cIAP2 [59] exhibits E3 ligase activity in homodimerized form only. Some RING E3s work as part of a large multi-subunit complex. For example, CRLs are large multi-subunit complexes that can ubiquitinate 300 different substrate receptors in humans, composed of a RING E3 (RBX1 or RBX2), a cullin protein (CUL1, CUL2, CUL3, CUL4A/4B, CUL5, or CUL7), and a protein substrate receptor [60,61]. U-box proteins are also classified into RING E3s because they use almost the same ubiquitin transfer mechanism, and the structure resembles the RING domain, though they lack zinc ions [62]. E2 can transfer ubiquitin from E2~ubiquitin to an ε-amino group of a substrate without an E3, but the process is inefficient. Other studies have shown that several E2~ubiquitin conjugates are not reactive, because they tend to have various inactive conformations. RING E3 can promote a population shift toward closed conformations, resulting in the efficient stimulation of the transfer activity of E2 (Figure 2A). The detailed mechanism has not been fully revealed yet [55].

Structural studies of the UbcH5 family E2s have revealed that a ubiquitin of E2 is shifted proximal to the RING domain by binding with RING E3. The RING domain binds both E2 and the Ile36 surface of ubiquitin that contacts α2 of E2 (Figure 2B). The C-terminal tail of ubiquitin is positioned to a favorite site for catalysis where an E2~ubiquitin thioester is attacked by an incoming substrate Lys.

Some RING E3s have an additional ubiquitin-binding component that enhances enzymatic activity by stabilizing E2~ubiquitin [63,64,65]. For example, the phosphate moiety of phosphor-Tyr36 of CBL-L forms a hydrogen bond with the Thr9 of ubiquitin, and loops adjacent to the RING domain of RNF38 contact the Thr9-containing surface of ubiquitin. In ARK2C, RING E3 is required for two ubiquitin-binding to exert transfer activity; one ubiquitin is located on the same surface of the E2-binding surface and another one binds to the opposite surface of RING E3 [66]. In the dimeric RING E3s, RNF4 and BIRC7, two domains cooperatively recognize ubiquitin: one subunit and a C-terminal tail of another subunit interact with the Gly35-containing surface of ubiquitin. On the other hand, TRIM25 uses a different interface for ubiquitin recognition: the TRIM25~UBE2D1 (UbcH5a)~ubiquitin complex structure revealed that the N-terminal helix of one subunit and C-terminal helix of another subunit make a hydrophilic interaction with the Gly35-containing surface of ubiquitin. In addition to E2-E3 interactions, several E3s harbor an additional domain for interacting with the backside surface of E2 that enhances the RING E3-E2 affinity but affects activity disparately [67,68,69,70]. Some E2s, such as RAD6 and the UbcH5, bind to ubiquitin on the backside surface of E2 to promote processive polyubiquitin chain formation [69,71,72]. 

These pieces of evidence indicated that additional components, domains, and molecules have distinct roles. Further structural and biochemical studies including those molecules are required for understanding RING E3-mediated ubiquitylation [55]. 

#### 3.3.3. HECT

There are 28 HECT E3s in humans [73]. The HECT E3s consist of an N-terminal substrate-binding domain and a C-terminal HECT domain. C-terminal HECT is a domain consisting of ~350 amino acids (Figure 3A). It was first described in human papillomavirus (HPV) E6-associated protein (E6AP) 5 [73]. HECT E3s are divided into three groups based on their N-terminal domain: NEDD4 family, HERC family, and HECTs with other protein–protein interaction domains. The HECT domain itself is divided into two lobes that are connected by a flexible hinge loop. The N-terminal lobe (N-lobe) binds to E2~ubiquitin, and the C-terminal lobe (C-lobe) has the catalytic cysteine residue [74]. The flexible hinge enables the lobes to rotate, leading to ubiquitin transfer reaction [75]. After the binding of E2~ubiquitin to the N-lobe, ubiquitin is transferred from E2 to the catalytic cysteine on the C-lobe. Then, the HECT~ubiquitin is juxtaposed with the substrate lysine residue that is ubiquitinated. Earlier structural studies indicated that conformational changes are required for the E2-E3 transthiolation reaction because the distances between E2 and HECT E3 are too long to achieve transfer reaction in the reported structures [74,75,76]. The crystal structure of NEDD4L in complex with UbcH5b~ubiquitin revealed that a rotation about the hinge is involved in positioning the catalytic cysteine of the C-lobe adjacent to the UBE2D2 (UbcH5b)~ubiquitin linkage [77]. Based on the NEDD4L structure, a transthiolation reaction model is proposed. The N-lobe initially recruits E2~ubiquitin, and upon rotation about the hinge, the C-lobe binds to ubiquitin and juxtaposes both catalytic cysteines to promote HECT E3~ubiquitin formation. However, the C-lobe residues are not conserved in all HECT E3s. Therefore, further studies are required for elucidating the transthiolation mechanism of other HECT E3s. The NEDD4~ubiquitin structure revealed that the interaction between ubiquitin and the C-lobe is similar to what has been observed for the primed ubiquitin in the RING E3-E2~ubiquitin complex, suggesting that RING and HECT E3s have the common thioester-activating mechanism. The Rsp5~ubiquitin–Sna3 complex structure showed a mechanism of how HECT E3s transfer ubiquitin to the substrate; the E3~ubiquitin thioester in HECT is juxtaposed with a substrate lysine. The C-lobe undergoes a 130° rotation about the flexible linker relative to the conformation in the NEDD4L-UbcH5b~ubiquitin and NEDD4~ubiquitin complexes. The N-lobe interacts with the C-lobe to stabilize the conformation. Phe806 of the C-lobe of Rsp5 is accommodated in the hydrophobic pocket of the N-lobe. Mutation analysis revealed that this hydrophobic interaction is required for locating the two HECT domain lobes in an orientation suitable for substrate ubiquitylation [78]. The amino acid composition of the N-lobe pocket is conserved in the NEDD4 E3s, although the amino acid composition is not conserved in other HECT E3s. This proposed mechanism seems to be conserved among HECT E3s. Unfortunately, the Rsp5~ubiquitin-Sna3 structure does not capture a substrate lysine poised for ligation. Further structural studies are required for elucidating the mechanism of how HECT E3s transfer ubiquitin to a substrate.

#### 3.3.4. Ring-between-Ring

The 14 E3s harboring RBR were identified in humans. All have a RING1-IBR-RING2 motif [55] (Figure 3A). Among RBR E3s, PARKIN, HHARI, and HOPI are well studied. RBR E3s are distinct from RING E3s because the studies of HHARI and PARKIN revealed that RBR E3s form a thioester intermediate with the C-terminal of ubiquitin in a HECT E3-like manner [55]. The RING1 domain recruits E2~ubiquitin and then transfers the ubiquitin to the catalytic cysteine of the RING2. Structural studies have revealed that only RING1 has a cross-braced architecture, which is the typical RING domain. Both IBR and RING2 regions have two zinc ions in their domain. The arrangement of each domain of the RBR is distinct among PARKIN, HHARI, and HOIP [55]. It is thought that the interaction between the RING1 and E2s is similar to those of canonical RING domains. As the RING1 harbors a hydrophobic core for interacting with the L1 and L2 loops of E2s, however, the RING1 domain does not have the linchpin arginine conserved in RING E3s, and RING1 alone cannot promote ubiquitin transfer [79,80]. The activation mechanism of RING1 should be different from the mechanism of RING E3s. On the other hand, these pieces of evidence suggest that RBR E3s transfer ubiquitin to a substrate by a mechanism such as HECT E3s. The HOIP-UbcH5b~ubiquitin structure revealed how ubiquitin is transferred to RING2 [65]. RING1, RING2, and IBR catch a UbcH5b~ubiquitin with multiple interactions. These interactions enable RING2 to interact with UbcH5b, resulting in the conformation stabilizing the C-terminal tail of ubiquitin in an extended conformation. The UbcH5 and the catalytic cysteine of RING2 are juxtaposed for leading ubiquitin transfer. PARKIN and HHARI also have two helices between RING1 and IBR, suggesting that these proteins share a similar mechanism of E2~ubiquitin binding. HOIP catalyzes a linear ubiquitin chain formation. HOIP has a linear ubiquitin chain-determining domain that facilitates binding and orients the acceptor ubiquitin. The HOIP-ubiquitin complex structure revealed how HOIP transfers a doner ubiquitin to an acceptor one [81]. The RING2 and the linear ubiquitin chain-determining domain bind to an acceptor ubiquitin. The C-terminal of a doner ubiquitin is located next to the active site cysteine of RING2. The C-terminal tail of the donor ubiquitin lies along a hydrophobic groove surrounding the catalytic cysteine in the RING2 domain. This binding stabilizes the tail of ubiquitin in an extended conformation. The RING2 of PARKIN and HHARI also have hydrophobic grooves, suggesting that RBR E3s having a hydrophobic groove may similarly transfer a ubiquitin, though they use a different domain outside the RBR motif for recognizing the substrate. Presumably, HOIP catalyzes the ubiquitination reaction identically. The mechanism transferring ubiquitin to a substrate by RBR E3s has not been revealed yet. Unfortunately, the Rsp5~ubiquitin–Sna3 structure does not capture a substrate lysine poised for ligation. Further structural studies are required for elucidating the mechanism of how HECT E3s transfer ubiquitin to a substrate. This proposed mechanism seems to be conserved among HECT E3s. Unfortunately, the Rsp5~ubiquitin–Sna3 structure does not capture a substrate lysine poised for ligation. Further structural studies are required for elucidating the mechanism of how HECT E3s transfer ubiquitin to a substrate.

#### 3.3.5. Atypical E3 Ligases of Ubiquitin-Like Proteins

Almost all E3 ligases are categorized into three types. However, some E3 ligases for Ubl are considered to be atypical E3 ligases. RanBP2 is a SUMO E3 ligase whose catalytic domain exists in the IR1-M-IR2 fragment [82]. RanBP2 known as Nup35 is one of the nuclear pore complex proteins. RanBP2 harbors multiple domains that interact with nuclear transport receptors, the GTPase Ran, Ubc9, and SUMOylated GTPase-activating protein RanGAP1 [13,83,84,85,86]. A 33 kDa fragment called IR1-M-IR2 of RanBP2 has E3 and, of interest, this fragment is almost unstructured. The RanBP2/UBC9/SUMO−RanGAP1 structure is considered a product complex after conjugation [82]. The complex revealed that the IR1-M domain of the RanBP2 domain contacts with the E2 UBC9 backside, and the SUMO-interacting motif binds the donor SUMO to position it in a closed conformation such as a RING-mediated E2~Ubl activation [42] (Figure 3B). RanBP2 is the first example for E3 ligase that is neither a HECT type nor a Ring finger type. ZF451 is also a SUMO E3 ligase harboring two SUMO-interacting motifs and C2H2-type Zinc finger motifs. The ZNF451/UBC9/SUMO−RanGAP1 complex structure revealed that ZNF451 uses an N-terminal SUMO-interacting motif to maintain the donor SUMO in a closed conformation, and the C-terminal SUMO-interacting motif to engage the second SUMO molecule that is bound on the backside of E2 UBC9 [87] (Figure 3B). The biochemical study revealed that the tandem-SIM region is sufficient to extend a backside-anchored SUMO chain, whereas efficient chain initiation (E3 ligase activity) requires a zinc-finger to recruit the initial acceptor SUMO [88]. The structural basis has not been unveiled yet.

Ubiquitin-fold modifier 1 (UFM1) is also one of the ubiquitin-like proteins. UFM1 is conjugated to its target proteins by a three-step enzymatic reaction. The UFM1-specific ligase 1 (UFL1) acts as the E3 to recognize its substrate, transfer, and ligate the UFM1 from E2 to the substrate. This process is called UFMylation, and the system is conserved in multicellular organisms. A UFM1 cascade is closely related to human diseases. UFM1 was covalently conjugated with C20orf116 [89,90]. UfL1 has no sequence homology to any other known E3s for ubiquitin and ubiquitin-like modifiers. However, structural studies have not been reported yet. The molecular mechanism remains unclear. 

The ATG12-ATG5 complex acts as an E3. The complex conjugates ubiquitin-like protein ATG8 to PE. ATG12-ATG E3 uses ATG3 as E2. Structural analyses of the ATG12−ATG5 complex revealed an extended interface between ATG12 and ATG5 that extends beyond the isopeptide linkage [91,92,93] (Figure 3B). This interface is required for the E3 activity of the ATG12-ATG5. ATG12-ATG5 has at least one more function in vivo. ATG12-ATG5 associates with ATG16 and the ATG-PE complex and forms a two-dimensional mesh organizing associated membranes. Dedicated studies have suggested that ATG12-ATG5 (E3) and ATG3(E2) activities are regulated through a series of protein-protein and protein-lipid interactions. The ATG12-ATG5 complex is an atypical E3 under multiple layers of regulation.

## 4. Newly Categorized E3 Ubiquitin Ligase PCAF_N

### 4.1. PCAF_N Domain

PCAF_N, previously known as the PCAF homology domain (PCAF-HD), is a newly categorized E3 ligase, as described below. PCAF_N is first identified as an N-terminal conserved domain between General control non-derepressible 5 (GCN5, KATA2A) and PCAF (P300/CBP-associated factor, KATA2B), both of which are well known as histone acetyltransferases. The PCAF and GCN5 are incorporated into SAGA (Spt-Ada-Gcn5-Acetyltransferase), ATAC (Ada-Two-A-Containing), TFTC (TBT-free-TAF complex), or PCAF complex [94]. These complexes are coactivators that are important for transcriptional activation by modifying chromatin. Both PCAF and GCN5 harbor the same domain architecture containing the PCAF_N domain, an acetyltransferase (AT) domain, and a bromodomain. While most of the metazoan genomes code the GCN5 gene, vertebrates encode PCAF [94]. The N-terminal PCAF_N domain seems to be metazoan-specific, and the AT domain and bromodomain in the C-terminus are highly homologous to yeast GCN5 (Figure 4A). In humans, GCN5 has two isoforms: the longer isoform harbors the PCAF_N domain but the shorter one lacks it (Figure 3A). In contrast, PCAF does not have a shorter isoform. 

### 4.2. PCAF_N Domain as a Ubiquitin E3 Ligase

It has been demonstrated that PCAF acts as an E3 ligase targeting human Hdm2, human Gli1, and human CIITA and promotes self-ubiquitination [84,85,86]. Although the longer isoform of GCN5 is thought to have E3 ubiquitin ligase activity, the experimental evidence has not been provided for a long time. Recent work has demonstrated that auto-ubiquitination activity was detected both in human and mouse GCN5 [18], but the actual substrate of GCN5 has not been identified yet. The E2 screening experiment using eight E2s (UbcH1, UbcH2, UbcH5a, UbcH5b, UbcH5c, UbcH7, UbcH8, and UbcH10) showed that UbcH5a and UbcH5c, in addition to UbcH5b, were likely to work as E2 enzymes for PCAF and GCN5, though UbcH5a, UbcH5b, and UbcH5c are highly homologous proteins [18]. A search of cognate E2s and the substrate protein of PCAF_N has just started.

PCAF_N exhibits a ubiquitin E3 ligase activity, indicating that the higher organism PCAF and GCN5 bear one more enzymatic activity in addition to an acetyltransferase activity. The relationship between E3 ligase activity and HAT activity has not been reported yet.

### 4.3. Crystal Structure of PCAF_N

An amino acid sequence analysis indicated that PCAF_N could not be categorized into three E3 protein families (Ring, HECT, and RBR). The recent structural work of PCAF_N demonstrated that this domain exhibits a unique structural motif distinct from the other three E3 ligases. 

To date, only one structure deposited into Protein Data Bank (PDB) is the 1.8 Å crystal structure of mouse GCN5 (Figure 4B). The crystal structure of PCAF_N of mGCN5 revealed that it folds into a single domain. E3 ligases usually have various biological assemblies to exert ubiquitination activity (Table 1). PCAF_N exists as a dimer in a crystal, raising the possibility that the PCAF_N works as a dimer. Size-exclusion chromatography with multi-angle light scattering (SEC-MALS) and size-exclusion chromatography with small-angle X-ray scattering (SEC-SAXS) experiments revealed that PACF_N exists as a monomer state in solution, suggesting that PCAF_N works as an E3 ligase in a monomer [18].

The 3D structural comparison using the DALI server [95] indicated that the overall structure is unique among the protein structures deposited into PDB. PCAF_N could be divided into three regions based on their characteristic structures. The N-terminal region (a. a. 83–159) coordinating two Zn ions was termed the Zn region. The second region was termed the connecting region that forms an anti-parallel coiled-coil structure and connects the former and the latter regions. The third region (a. a. 216–372), folding into an α-helix-rich structure, was termed the MORF4-related gene domain male-specific lethal3-like (MSL3-like) domain (MSL3; PDB ID: 2y0n) [96], because the 3D structure comparison analysis by the program DALI indicated that the C-terminal region is homologous of MSL3, although the sequence identity between them is low (approximately 22%).

The most intriguing structural feature of PCAF_N is the Zn region. Amino acid sequence analysis could not predict that the PCAF_N domain can coordinate Zn ions, indicating that they do not bear a typical Zn binding motif such as a RING finger. X-ray crystallography is a useful method for identifying a metal ion. The XAFS analysis and anomalous Fourier map could clearly identify Zn ions that corresponded to the strong electron densities observed in the N-terminal region of PCAF. Strikingly, the coordination manner is unique. The Zn region has a binuclear Zn-coordination structure (Zn_2_Cys_5_His_2_) (Figure 4B,C). The two Zn ions coordinate with seven residues (Cys107, Cys113, Cys115, Cys142, Cys145, His147, and His151). The sulfur atom of Cys145 coordinates both Zn ions (Figure 4C). All the Zn-coordinating residues of the PCAF_N domain are highly conserved among the proteins harboring the PCAF_N domain (Figure 4C). E3 ligase RAG1 has three Zn binding sites, and one of them is a similar Zn coordination (Zn_2_Cys_5_His_2_ binuclear cluster) [97]. It should be noted that the binuclear Zn coordination of the RAG1 is outside of the E3 ligase region, indicating that there is no provided evidence for a relationship between E3 ligase activity and this binuclear Zn coordination. The HMM logo shows that the residues coordinating Zn ions and Trp residue are highly conserved in PCAF_N family proteins (Figure 4D). The Trp is W118 in the PCAF_N domain of mGCN5. W118 forms hydrogen bonding with the main chain carbonyl oxygens of E140 and G102, indicating that this highly conserved Trp residue stabilizes the structure of the Zn region (Figure 4E).

This compact two-Zn-ion coordination resembles the RING domain, but the Zn coordination pattern and ternary structure are unique compared to the RING domain. The lack of a Zn region lost the ubiquitin E3 activity of mGCN5 [18], indicating that the Zn region is responsible for exerting E3 activity. However, it has not been unveiled whether the Zn region only has ubiquitin E3 activity as with a RING E3.

All E3 ligases harbor an E2~ubiquitin-binding domain. In E2-RING structures, RINGs exhibit a common mode of interaction with E2 (Figure 2B). In the PCAF_N domain, two loops coordinating Zn forms a narrow groove together with α-helices (α2 and α3), although its ternary structure and coordination pattern of the PCAF_N domain were different from those of RING E3 ligases. To identify the binding surface, further studies are necessary. RING E3s directly transfer ubiquitin to a substrate. HECT and RBR E3 ligases transfer ubiquitin to their catalytic cysteine and then transfer the ubiquitin to a substrate; HECT and RBR E3 ligases harbor the catalytic cysteine that receives ubiquitin from E2-ubiquitin. The GCN5 PCAF_N domain and PCAF have three conserved cysteines except for the Zn coordinating cysteines. Those cysteines are part of a hydrophobic core, indicating that these cysteines are unable to receive a ubiquitin.

The MSL3-like domain enhances the activity of HAT MOF by interacting with MSL1 [96]. The MRG domain is enabled to interact with diverse groups of proteins. RING E3 should recruit both E2 and the substrate protein. The MSL3-like domain might work as a substrate-binding site.

The property of the MRG domain interacting with diverse proteins might provide PCAF_N with the ability to transduce specific and unknown signals in the cell. Interestingly, GCN5 and PCAF have a bifunctional enzymatic activity that is ubiquitin E3 ligase activity and acetyltransferase activity, and both functions target Lys residues as a substrate. This additional function of GCN5 and PCAF conserved in vertebrates should provide a specific signal in higher organisms. To address these issues, further structural and functional studies are required.

### 4.4. PCAF_N Family

The Pfarm database indicates that the PCAF_N domain is identified in 393 species of Eukaryota. The PCAF_N family includes 1006 sequences and 26 architectures. About half of the architectures belong to the histone acetyltransferase protein, but PCAF_N is also spread in proteins with other functions.

The proteins having a partial region of the PCAF_N were also found. The structural study showed that the Zn region is required for exerting E3 ligase activity, but the essential region as an E3 ligase has not been identified. The proteins harboring the PCAF_N domain may function as an E3 ligase, but some proteins lacking a part of PCAF_N would lose their function as an E3 ligase (Figure 5). Further functional analysis and structural studies would shed light on the diverse unknown function of the PCAF_N family.

## 5. Future Perspective

Extensive functional and structural studies have uncovered ubiquitination reaction mechanisms. E3 ligases show diverse molecular sizes and domain architecture. Both the small RING domain and large CRLs complex catalyze the ubiquitination reaction. This structural diversity is responsible for various ubiquitination signals in the cell. Moreover, multidomain and/or oligomerization of E3 ligases appear to be key. A newly categorized PCAF_N domain is also incorporated in the multidomain protein. Current structural studies have encompassed only a part of all E3s; future studies are required to uncover the functions and mechanisms of other unknown and/or uncharacterized E3s, leading to a new avenue for understanding unveiled ubiquitin signals and developing a new drug and a good tool for PROTAC.

## Figures and Tables

**Figure 2 molecules-26-06682-f002:**
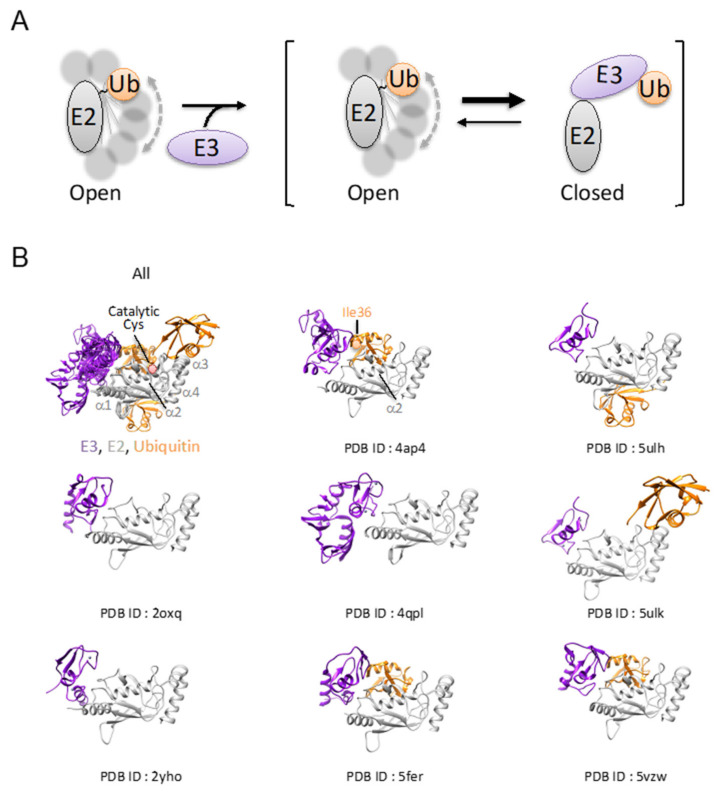
Recognition of E2 by RING E3. (**A**) Schematic diagram of E2~Ub activation mechanism by RING E3. The structure of E2~ubiquitin prefers open conformations in which a ubiquitin molecule moves dynamically. RING E3 promotes a population shift toward closed conformations to stimulate the transfer activity of E2. (**B**) The crystal structures of the RING E3-UbcH5 complex. Ubiquitin, E2, and RING E3 are shown in a ribbon diagram and colored in orange, gray, and purple, respectively. PDB ID is shown under each structure. The position of catalytic cysteine is indicated as a pink circle. The Ile36 located on the ubiquitin surface contacting α2 of E2 is indicated as an orange circle.

**Figure 3 molecules-26-06682-f003:**
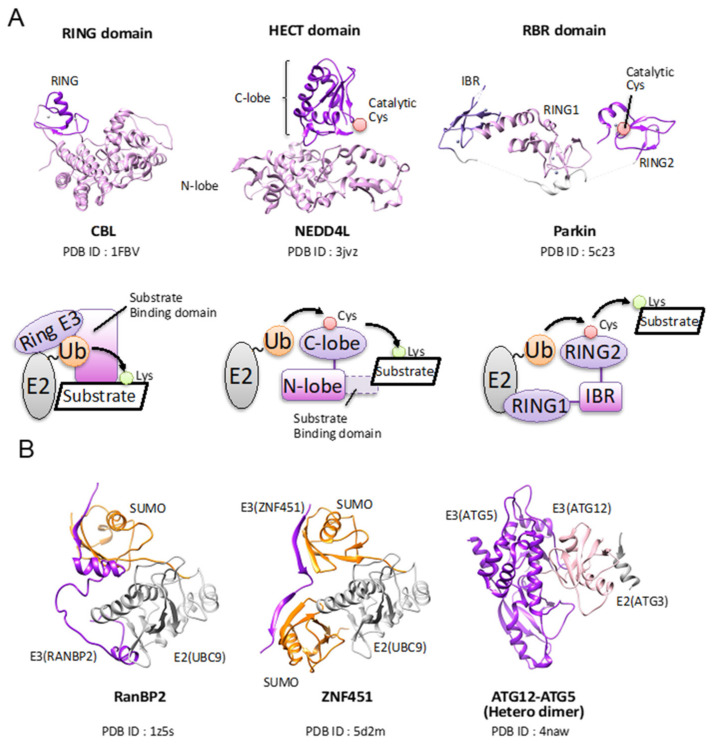
Structures of classical and atypical E3 ligases. (**A**) The crystal structures of the RING E3, HECT E3, and RBR E3 domain are drawn in a ribbon diagram. The molecular name and PDB ID are shown under each structure. In the RING E3 structure, the RING domain is colored in purple, and the remaining structure is colored in pink. In HECT E3, N-lobe and C-lobe are colored in pink and purple, respectively. In RBR E3, RING1, IBR, and RING2 are colored in pink, pale purple, and purple, respectively. The linker region between IBR and RING1 is colored in gray. A pink circle indicates the position of catalytic cysteine. The schematic diagram of the ubiquitination mechanism of each E3 is drawn. (**B**) The crystal structures of atypical E3 ligase. The molecular name and PDB ID are shown under each structure. The structure of Ubl, E2, and E3 molecules are drawn in a ribbon diagram and colored in orange, gray, and purple, respectively.

**Figure 4 molecules-26-06682-f004:**
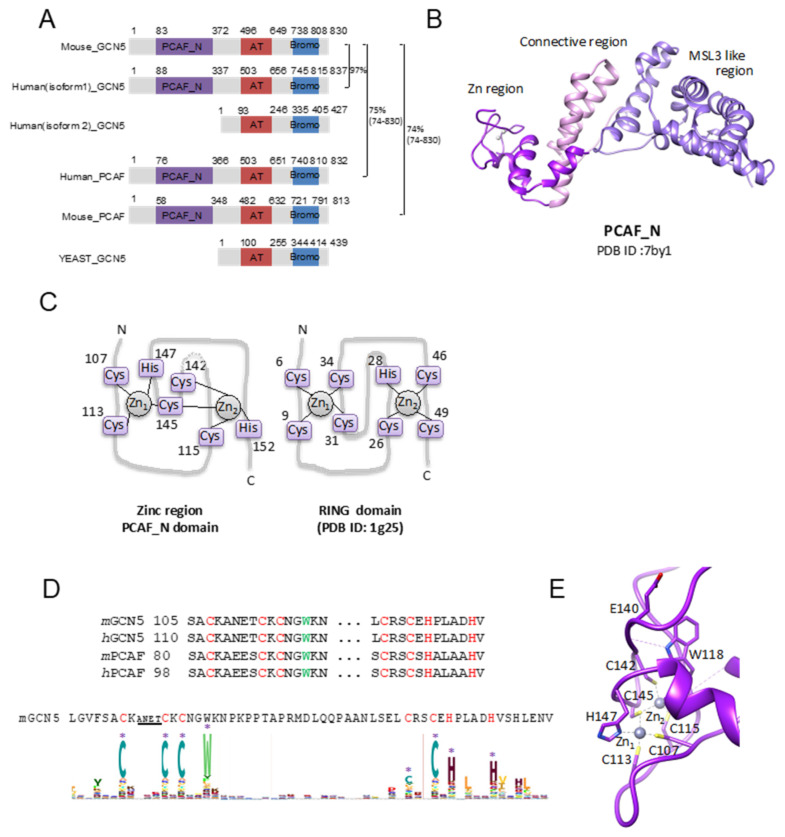
Structure of PCAF_N domain. (**A**) Domain architectures of GCN5 and PCAF. The PCAF_N domain, the acetyltransferase domain (AT), and the bromo domain (Bromo) are indicated as a box and colored in purple, red, and blue, respectively. The amino acid sequence identities (%) are indicated on the right in which the number in parentheses indicates the amino acid region using sequence alignment. (**B**) The crystal structure of PCAF_N is drawn in a ribbon diagram in which the Zn region, the connective region, and the MSL-like region are colored in purple, pink, and pale purple, respectively. The PDB ID is indicated under the ribbon diagram. (**C**) The topology diagrams of the Zn-coordinating manner of the Ring region of the PCAF_N domain and Ring domain. The Zn-coordinating residues are indicated as the purple box. The PDB ID of the RING domain is shown. (**D**) The result of HMM logo in Pfarm (edited) is shown. The region around the residues coordinating Zn ions is shown. The seven ligand residues are indicated by an asterisk (*). The font size in HMM logo analysis shows the conservation of amino acids within the multiple sequence alignment. The amino acid drawn in a bigger size indicates that the amino acid is highly conserved among the protein family. GCN5 and PCAF have one amino acid insertion between the first and second cysteine residues. The underline indicates the corresponding sequence. (**E**) The structure around the Zn binding site of PCAF_N. Two Zn ions are drawn in the sphere model, and the coordination manner is indicated by the dashed black line. The Zn coordinating residues and highly conserved Trp residue (W118 in PCAF_N domain of mGCN5) are drawn in the stick model. W118 forms hydrogen bonding with the main chain carbonyl oxygens of E140 and G102. The two hydrogen bonds are indicated by a dashed purple line.

**Figure 5 molecules-26-06682-f005:**
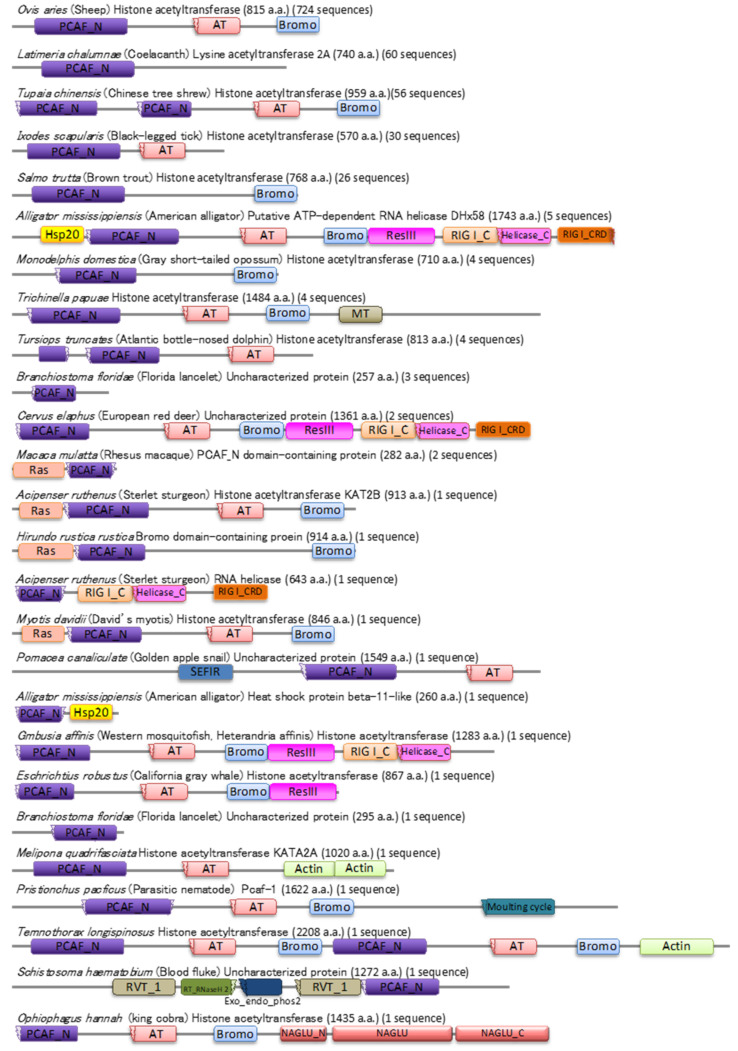
The domain architectures of PACF_N family proteins. The structured domains are shown. The names (species and proteins) are indicated on each domain architecture. The value in the first parenthesis is the number of amino acid residues of the presented protein. The value in the second parenthesis is the number of protein sequences harboring the same domain architecture.

**Table 1 molecules-26-06682-t001:** Examples of biological assembly of ubiquitin E3s.

Examples of Ubiquitin E3s		
Name	Biological Assembly	Protein (PDB ID)
RING E3	Monomer	CBL (1fb), RNF38 (4v3l), CNOT4(1ur6), ARK2C (5d0m, 5d0k)
	Homodimer	RNF4 (4ppe, 4ap4), cIAP2(3eb6), BIRC7 (4auq),TRAF6 (3hcs, 5vo0)
	Heterodimer	MDM2-MDMX (2vje), BRCA1-BARD1 (1jm7),RNF2-BMI1 (2ckl)TRAF6:TRAF5 (7l3l)
	Homodimer, heterodimer, or oligomers	TRIM family proteins (TRIM65(7jl2), TRIM5a(4tkp), TRIM28(6i9h), TRIM32(5fey))
	Component of multi-subunit	APC/C (APC11 (4r2y, 5jg6, 5lt9)), CRL (RBX1 (4f52, 1ldk, 4a0l), RBX2 (7oni))
	(U-box)	
	Monomer	UBE4B known as UFD2 (2qj0)
	Homodimer	PRP19(2bay), CHIP (2c2v)
HECT E3	Monomer or oligomer	SMURF1 (1zvd), NEDD4L (3jvz), WWPI (1nd7), E6AP(1c4z)
RBR	Monomer or oligomer	PARKIN (5c23), HHARI (5tte), HOIP(4ljo)
PCAF_N	Monomer or component of multi-subunit	GCN5 (7by1)
Atypical	Monomer?	ZBF451(5d2m)
	Heterodimer	ATG12-ATG5(4naw)
	Component of multi-subunit	RanBP2(1z5s)

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
