# Peer review of "Structural Diversity of Ubiquitin E3 Ligase"

_molecules, 2021, doi:10.3390/molecules26216682_

Round 1

Reviewer 1 Report

The author has written a comprehensive overview of the structure and function of ubiquitin E3 ligase, and the manuscript is expected to contribute to related research fields. However, a lot of text errors are found in the manuscript, and the current version of the manuscript is difficult to support for publication. For publication, the following should be improved. 1. Authors should review the notation related to the reference. e.g. line 34-35: degradation [6] <period after reference> line 41: [10]. [11,12] <reference merge> line 72: polymer[17] <add space between letter and reference> 2. Authors should review the notation in the figure caption. e.g. line 84: (A. line 87: bond. . B. 3. The name of the strain must be written in italics. e.g. line 109: Saccharomyces cerevisiae 4. In Figure 2. we need to center the text. 5. Authors must use consistence expressions. e.g. "PDB id" or "PDB ID" "Fig. X" or "FIG. X" 6. It is difficult to distinguish the bottom sequence of Figure 4D. 7. Authors need to increase the text resolution in Figure 5. 8. Table (line 495~) should not be divided into sections, but should be placed in an appropriate location and the character style should be unified.

Reviewer 2 Report

Shimizu and Toma-Fukai summarize the recent progress towards structural understanding of E3 ligases. A newly categorized E3-PCAF_N is also discussed. The review is well-written with current knowledge, comments of present approaches, and possible future advances. I am supportive of the publication of this manuscript with a few clarifications listed below.

  1. In Figure 1. A, it’d be beneficial to show the actual thioester bond between E2 and ubiquitin, isopeptide bond between E3 and substrate, from a chemistry point of view.
  2. In Figure 4. D, the picture below the sequences is too small to see.
  3. What’s the function of the table at the end of the manuscript? There’s only one place (3-4-2) mentioning it. The table is a very good summary and more discussion would be good.

Reviewer 3 Report

Comments for the Authors

The authors have summarized in this Review the most recent developments in the study of the Ubiquitin (Ub)-E3 ligases. In particular, the aim of the paper is, within reason, to systematically review the current knowledge on structural diversity of such peculiar and complex enzymes.

In the last decades, proteasome pathway and its components -Ub-activating enzyme E1, Ub-binding enzyme E2 and Ub-ligase E3- have been worth considering as a novel strategy that targets disease-related proteins for degradation. For this reason a review on the subject might be of great -interest.

The manuscript appears to be an extensive, well-written and well-conceived study reviewing the available knowledge on the matter, endowed with clarity, accuracy, consistency and balance. Particular emphasis has been devoted to structural features of the different class belonging to E3 ligases. In addition to molecular details regarding the well-known families of proteins, the Authors have included an in depth-analysis of PCAF_N, a newly categorized E3 ligase.

Here, some suggestions for improving the manuscript:

  • A less precise writing in terms of both syntax and correctness of the English language hinders the Introduction, compared to the other paragraphs of the manuscript. Authors should improve the final output
  • I suggest moving 3-1 paragraph (Structure of Ubiquitin) before the actual 2-1 paragraph (Ubiquitination reaction).
  • Proteolysis targeting chimeric (PROTAC) technology is an effective endogenous protein degradation tool that can ubiquitinate the target proteins through the ubiquitin-proteasome system (UPS) to achieve an effect –as an example- on tumor growth. A number of literature studies on PROTAC technology have proved an insight into the feasibility of PROTAC technology to degrade target proteins. Additionally, the first oral PROTACs (ARV-110 and ARV-471) have shown encouraging results in clinical trials for prostate and breast cancer treatment. Thus, perhaps it is a leap to call "new" a technology that was first described two decades ago (see the Introduction).
